# Medical Vision Generalist: Unifying Medical Imaging Tasks in Context

## Abstract

This study presents Medical Vision Generalist (MVG), the first foundation model capable of handling various medical imaging tasks—such as cross-modal synthesis, image segmentation, denoising, and inpainting—within a unified image-to-image generation framework. Specifically, MVG employs an in-context generation strategy that standardizes the handling of inputs and outputs as images. By treating these tasks as an image generation process conditioned on prompt image-label pairs and input images, this approach enables a flexible unification of various tasks, even those spanning different modalities and datasets. To capitalize on both local and global context, we design a hybrid method combining masked image modeling with autoregressive training for conditional image generation. This hybrid approach yields the most robust performance across all involved medical imaging tasks. To rigorously evaluate MVG's capabilities, we curated the first comprehensive generalist medical vision benchmark, comprising 13 datasets and spanning four imaging modalities (CT, MRI, X-ray, and micro-ultrasound). Our results consistently establish MVG's superior performance, outperforming existing vision generalists, such as Painter and LVM. Furthermore, MVG exhibits strong scalability, with its performance demonstrably improving when trained on a more diverse set of tasks, and can be effectively adapted to unseen datasets with only minimal task-specific samples. The code will be available soon.

## 1 Introduction

The precise interpretation of medical images is imperative for timely disease detection, diagnosis, and treatment Cheng et al. (2022b); De Fauw et al. (2018). Deep-learning based models have emerged as powerful tools in medical image analysis, tackling various challenges spanning from segmenting specific anatomical structures Ji et al. (2022); Luo et al. (2021); Fu et al. (2021), localizing single organ diseases Zhu et al. (2019); Zhao et al. (2021); Huo et al. (2020); Cheng et al. (2022a); Ardila et al. (2019); Kim et al. (2022); Heller et al. (2021), to cross-modality image synthesis on brain MRI Xie et al. (2023); Li et al. (2023); Dayarathna et al. (2023); Zhu et al. (2023). However, these models, often referred to as specialist models, are typically customized for specific tasks, modalities, or anatomical regions. While this specialization often results in exceptional performance in certain contexts, it can lead to a severe performance drop when applied to new tasks or when tasked with training multi-domain data.

To address this challenge, recently, there has been a partial shift of research focus in developing generalist medical AI models Moor et al. (2023); Tu et al. (2023), which necessitate only a single training phase but are capable of wide application across a diverse array of medical tasks. Specifically, these generalist frameworks unify input and output spaces, allowing straightforward adaptation to various tasks through user-provided prompts. While existing generalist medical AI models like MedSAM have demonstrated impressive performance Ma et al. (2024); Zhang et al. (2023); Butoi et al. (2023), their applicability in medical visual tasks remains limited (*e.g.*, to segmentation tasks only). A unified, truly generalist vision model capable of addressing a vast array of medical imaging tasks remains a critical missing piece in the current medical research landscape.

Motivated by the remarkable success of in-context learning in natural language processing Brown et al. (2020a); OpenAI (2023) and computer vision Bai et al. (2023); Wang et al. (2023b;a), we hereby propose Medical Vision Generalist (MVG), the **first** generalist model in the medical imaging domain.

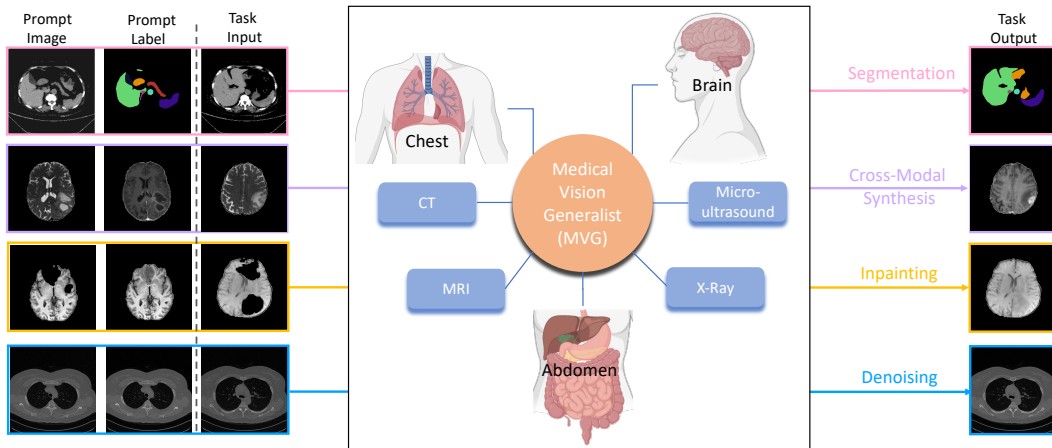

Figure 1: **Medical Vision Generalist** enables a single model to be capable of performing **four** types of medical vision tasks on images in **four** medical imaging modalities of multiple body regions.

Specifically, MVG leverages an in-context learning framework to unify a set of medical imaging tasks, including cross-modal synthesis, denoising, segmentation, and inpainting across modalities like CT, MRI, X-ray, and Micro-ultrasound. In contrast to prior task- and data-specific medical AI models, MVG offers adaptability to new data with minimal labeled samples, eliminating the need for retraining. To achieve this, MVG first standardizes the input/output space using in-context coloring, which maps various tasks into a single-channel coloring scheme. This removes the need for task-specific heads, thus regulating the model to learn exclusively from prompts. Subsequently, tasks are unified through conditional image generation, where MVG generates the output conditioned on both the task prompt and a sample image.

To capture both local and global context, we devise a hybrid strategy that combines masked image modeling and autoregressive training for conditional image generation. The former involves concatenating prompt images, labels, task inputs, and labels, followed by random masking; the latter constructs prompt image-label pairs, task inputs, and labels as long visual sentences. During inference, MVG conditions predictions on the prompts selected from locations closely matching the task images, ensuring contextual relevance and guidance that enhances output quality and consistency.

Furthermore, we have curated the first unified medical imaging benchmark, encompassing 13 datasets spanning a range of human anatomies (*e.g.*, abdomen, pelvis, brain, chest) and modalities (*e.g.*, CT, MRI, X-ray, micro-ultrasound). This new benchmark enables a comprehensive assessment of our MVG models. Experimental results demonstrate the effectiveness of our MVG in performing various medical vision tasks with only one model. As illustrated in Figure 2, our MVG outperforms the previous generalist models by a large margin. For instance, our MVG achieves 0.735 mIoU on all segmentation tasks and outperforms the previous best vision generalist by 0.123 mIoU. Furthermore, our MVG demonstrates two intriguing properties: 1) it scales well with multiple tasks and datasets, suggesting its potential to excel further as diverse datasets continue to emerge; and 2) it can efficiently generalize to new datasets, with only a few specific examples needed for each task.

## 2 RELATED WORK

**Medical Image Analysis.** In the field of medical image analysis, there have been key developments in deep-learning models for image segmentation. As the earliest success in this line of work, U-Net Ronneberger et al. (2015) uses an encoder-decoder architecture with skip-connection, revealing the great potential of deep networks. Following the line, nnUnet Isensee et al. (2021) further improves the model architecture and introduces bags of tricks, building a well-engineered general segmentation model. TransUnet Chen et al. (2021a) proposes to use pre-trained ViT for better feature extraction. Recent efforts in medical image analysis have produced remarkable models capable of performing a variety of tasks. Notable works include "One model to rule them all" Zhao et al. (2023), MedSAM Ma et al. (2024), and UniverSeg Butoi et al. (2023), which are designed to tackle unified medical segmentation tasks. UniverSeg adapts UNet Ronneberger et al. (2015) to intake in-context samples

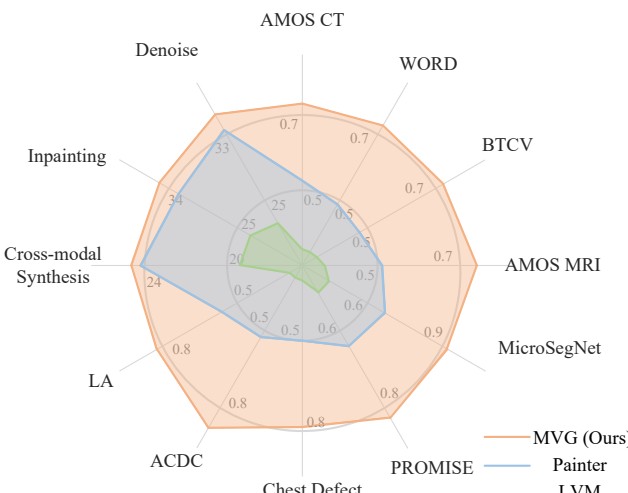

Figure 2: **Comparison with other generalists.** Our model achieves state-of-the-art performance on all involved medical vision tasks of five types.

to segment new data and tasks without further training. Besides, biomedGPT Zhang et al. (2023) proposes a unified generative model for bio-medical vision-language tasks. In this paper, we propose a novel paradigm to build a generalist model, which is capable of handling various medical vision tasks, including segmentation, inpainting, cross-modal synthesis, and denoising.

**Universal Models and In-Context Learning.** The advent of the universal Transformer architecture and its success in generative pretraining has inspired the development of universal models that tackle a wide range of computer vision tasks Chen et al. (2021b; 2022); Lu et al. (2022); Wang et al. (2022); Bai et al. (2023); Wang et al. (2023a). In-context learning is a novel few-shot learning paradigm that emerged in large language models and was first proposed by GPT-3 Brown et al. (2020b). Specifically, in-context learning enables one model to perform different tasks with only in-context examples as prompts. While the prompts for language models are mostly defined as a few sentences, in-context learning in other domains is still in an early exploration stage. As one of the earliest works, Flamingo Alayrac et al. (2022) extends the modality of in-context learning with language instructions and sequences of images and videos. Perceiver-IO Jaegle et al. (2021) uses the Transformer architecture for a general-purpose model that handles data from arbitrary settings like natural language, visual understanding, multi-modal reasoning, and StarCraft II. AD Laskin et al. (2022) introduces in-context learning to reinforcement learning with algorithm distillation. DPT Lee et al. (2024) provides a sample-efficient RL algorithm with strong in-context decision-making. In this paper, we use a sequence of paired medical images to build a vision model with in-context learning ability, unifying 13 medical tasks as a generation task.

## 3 METHOD

Unlike previous medical AI models, which are specific to one or a few predefined imaging tasks and produce a predetermined set of outputs, the proposed MVG aims to offer unprecedented flexibility across tasks, modalities, and datasets. The key idea is to unify medical imaging tasks, such as cross-modal synthesis, image segmentation, denoising, and inpainting, within an image-to-image generation framework.

### 3.1 TASKS

Our MVG is designed to address various medical imaging tasks, with a particular emphasis in this study on segmentation, cross-modal synthesis, inpainting, and denoising tasks for which well-represented public datasets are available. However, it is crucial to note that its design should be widely applicable to any image-to-image generation task.

**Segmentation.** Medical image segmentation, including CT, MRI, X-ray, and Micro-ultrasound segmentation, involves dividing an image obtained from these modalities into distinct segments to isolate regions of interest, such as organs or abnormalities. The input space for these tasks typically consists of images from CT, MRI, X-ray, or Micro-ultrasound scans. The output space is represented by a mask, where each value (excluding the background) in the mask corresponds to a different class or type of object, such as a liver or kidney.

**Cross-modal synthesis.** Cross-modal synthesis aims to generate images in one modality from images of another modality for the same subject, aiding in visualization and facilitating multi-modal medical image analysis. The input space and output space are different medical imaging modalities.

**Brain image inpainting.** In the context of brain image processing, inpainting refers to the process of synthesizing healthy brain tissue in regions affected by glioma, a type of brain tumor Kofler et al. (2023). Inpainting allows professionals to effectively utilize non-standard imaging protocols and directly apply brain parcellation tools to facilitate treatment planning. The input space is the corrupted brain MRI and the output space is the corresponding brain MRI restoring the affected regions to a normal state.

**Denoising.** Denoising aims to reconstruct full-dose CT images from low-dose CT images, allowing for reduced radiation doses during CT scans while preserving diagnostic image quality. The input space is the scanned CT image with low-dose radiation, while the output space is the corresponding image with full-dose radiation.

## 3.2 UNIFYING THE INPUT/OUTPUT SPACE

Assume an input image is denoted as $\mathbf{x} \in \mathcal{R}^{H \times W}$, the output could be a segmentation map, a synthesized brain image in the target modality, a restored normal brain MRI, or a full-dose CT image of the same size. To unify the output space of images across tasks, our MVG adopts a strategy beyond task-specific heads: mapping different tasks into a single-channel coloring scheme, inspired by Wang et al. (2023a;b). Specifically, we explore three different in-context coloring methods for segmentation that circumvent reliance on label values, including binary, pre-defined, and random colorization.

**Binary colorization.** We break down the problem of segmenting multiple classes into individual binary segmentation tasks, each focusing on separating one class from the background. Specifically, if a segmentation mask contains $N_k$ foreground classes, we simply split it to $N_k$ binary masks. However, this requires multiple inferences when an image contains more than one foreground class.

**Pre-defined colorization.** In this approach, we allocate a predetermined unique color to each segmentation mask derived from diverse datasets. Suppose there are $K$ segmentation datasets, with each dataset containing $N_k$ classes. Consequently, the $n_{th}$ class of the $k_{th}$ dataset is assigned the value of $\sum_{i=1}^{k-1} i * N_i + n$. Note that different tasks may involve classes with identical semantics; for example, both the AMOS segmentation dataset Ji et al. (2022) and the Synapse dataset Landman et al. (2015) include the class "Liver". However, distinct colors are assigned to the same class across different tasks.

**Random colorization.** The use of pre-defined colors may restrict the adaptability and efficacy of MVG, as they can cause the model to focus on learning tasks based on the color of the prompt rather than the contextual information Wang et al. (2023b). To address this limitation, we build a set of colors and randomly sample colors for different semantics in one iteration but the same semantic in the prompt label and task label share the same color.

Except for medical image segmentation, the outputs of all other tasks in this study do not involve categorical values that need to be predicted. Therefore, we do not apply coloring for these tasks.

## 3.3 TASK UNIFICATION VIA CONDITIONAL IMAGE GENERATION

After standardizing the input and output space for all tasks as images of identical sizes, we construct the training input, including 1) the task prompt consisting of paired prompt images and prompt labels, and 2) the task input and its associated label. We then unify various medical imaging tasks within a conditional image-to-image generation framework using the task prompt as task specification. All

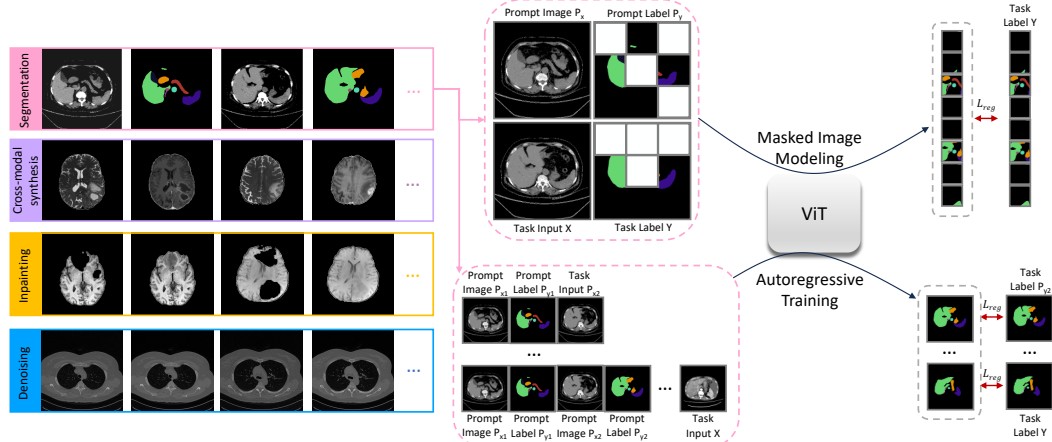

Figure 3: **Method overview. Left:** Four types of medical tasks (*i.e.*, segmentation, cross-modal synthesis, inpainting, and denoising) are unified as a universal image-to-image generation task with in-context learning. **Right:** We adopt mask image modeling and auto-regressive training for in-context generation.

the tasks are unified to generate the task label $Y \in \mathcal{R}^{H \times W}$ based on the condition including the task image $X \in \mathcal{R}^{H \times W}$, the prompt image $P_x \in \mathcal{R}^{H \times W}$, and the prompt label $P_y \in \mathcal{R}^{H \times W}$. Specifically, we use two conditional image generation frameworks: masked image modeling He et al. (2022) and autoregressive training Chen et al. (2020).

**Architecture Selection.** Following the same setting in He et al. (2022); Hua et al. (2022), we take vanilla ViT Dosovitskiy et al. (2020) as an encoder including a patch embedding layer and several Transformer blocks. The decoder is a simple prediction head with two convolution layers and takes four feature maps Li et al. (2022) from ViT as input.

**Mask Image Modeling.** During training, we form a square image by concatenating the prompt image (upper left) with its corresponding label (upper right), as well as the task image (lower left) with its associated label (lower right), as illustrated in Figure 3(a). We perform random masking on the square image and train ViT to reconstruct the masked region Wang et al. (2023a):

$$p(x) = \prod_{i=1}^{M} p(x_i | x_{x \notin x_M}, \theta). \tag{1}$$

where $x_M$ is the mask region, $x_{x \notin x_M}$ is the visible region, and $\theta$ denotes the model parameters. However, in practice, we observed that mask image modeling yields unsatisfactory results for medical image segmentation. We hypothesize that this may be attributed to the masking strategy's potential to compromise the preservation of global contextual information within individual images, such as the interplay among various abdominal organs. Furthermore, for small organs like the pancreas and the gallbladder, this masking approach can render them nearly invisible in prompts and makes prompts provide no in-context information about these organs. In contrast, for tasks like inpainting and denoising, masked image modeling excels, as these tasks prioritize refining local details over preserving global contextual information. To ensure the efficacy on medical image segmentation, we introduce an additional *auto-regressive* training, which preserves the global context within individual images, as shown below.

**Auto-Regressive Training.** In auto-regressive training, each image, including paired prompt images, prompt labels, task inputs, and associated labels, is treated as a single element in a sequential data structure. The model is fed with a partial sequence and trained to predict the next image in the sequence conditioning on the preceding ones.

Mathematically, let $P_{x_1}, P_{y_1}, ..., P_{x_n}, P_{y_n}, X, Y$ denote $n + 1$ pairs of images and labels. The first $n$ pairs serve as the task prompt, and the model learns to predict the task output $Y$ given the task

| Region | Dataset | Modality | #Training | #Testing | Task |
|--------|---------|----------|-----------|----------|------|
| Abdomen | AMOS Ji et al. (2022) | CT | 240 | 120 | Segmentation |
| Abdomen | WORD Luo et al. (2021) | CT | 100 | 20 | Segmentation |
| Abdomen | BTCV Iglesias & Sabuncu (2015) | CT | 21 | 9 | Segmentation |
| Abdomen | AMOS Ji et al. (2022) | MRI | 60 | 50 | Segmentation |
| Pelvis | MicroSegNet Jiang et al. (2024) | Micro-US | 55 | 20 | Segmentation |
| Pelvis | PROMISE Litjens et al. (2014) | MRI | 50 | 30 | Segmentation |
| Brain | BraTS-GLI Kazerooni et al. (2023) | MRI | 1251 | 219 | Cross-modal synthesis |
| Brain | BraTS-Local Kazerooni et al. (2023) | MRI | 1000 | 251 | Inpainting |
| Chest | Low dose McCollough et al. (2021) | CT | 200 | 59 | Denoising |
| Chest | Defect Detection Candemir & Antani (2019) | Xray | 15 | 6 | Segmentation |
| Chest | ACDC Bernard et al. (2018) | MRI | 100 | 50 | Segmentation |
| Chest | LA Chen et al. (2019) | MRI | 81 | 20 | Segmentation |
| Whole body | Deeplesion Yan et al. (2018) | CT | 25000 | 7120 | Detection |

Table 1: **Datasets overview.** Our MVG is trained and evaluated on 13 different datasets covering four major human body regions (*i.e.*, Abdomen, Pelvis, Brain, Chest). #Training/Testing refers to the number of samples for training and testing.

input $X$ and the prompt. This process iterates through each pair in the sequence. For each iteration, auto-regressive training is conducted with supervision solely on prompt labels and the task label:

$$S = [S_1, S_2, ..., S_{2n-1}, S_{2n}, S_{2n+1}, S_{2n+2}] = [P_{x_1}, P_{y_1}, ..., P_{x_n}, P_{y_n}, X, Y],$$

$$p(x) = \prod_{i=1}^{n+1} p(S_{2i}|S_1, ..., S_{2i-1}, \theta). \tag{2}$$

**Loss Function.** Any regression loss function like $l_1$ or $l_2$ can serve as the loss function of our MVG. Different from the $l_2$ loss function in masked image modeling He et al. (2022), we find the smooth $l_1$ performs best for MVG.

**Inference.** We first construct a sequence $S = [P_x, P_y, X, \hat{Y}]$, where prompts, the task image, and the desired output are concatenated together. MVG leverages the task prompt, composed of the prompt image $P_x$ and label $P_y$, for task specification, subsequently generating predictions by conditioning on both the task input $X$ and the task prompt. Since the task prompt is formulated as images, MVG demonstrates versatility in defining imaging tasks, capable of handling data sourced from diverse scanning machines, procedures, settings, or populations. For instance, if $P_x$ and $P_y$ represent an image and label extracted from the AMOS CT training set, respectively, MVG performs multi-organ segmentation on the image $X$ derived from the AMOS CT testing set, maintaining consistency within the dataset setting and guided by the provided context.

Different task prompts can yield varying results. In this study, we address this variability by selecting the prompt image from a location that closely matches the task image. Given the instance $X_{TE}$ which has $N_{TE}$ slices from the testing set, we randomly choose an instance $X_{TR}$ which has $N_{TR}$ slices and the corresponding label $Y_{TR}$ from the training set. For the $n_{th}$ slice of $X_{TE}$, we also choose the the floor$(\frac{n_{th}*N_{TR}}{N_{TE}})$ slice as the prompt.

## 4 EXPERIMENT

### 4.1 IMPLEMENTATION DETAILS

**Data.** As shown in Table 1, our model is developed on 13 different datasets including 2.5M training images covering four major human body regions (*i.e.*, Abdomen, Pelvis, Brain, Chest). Following the standard preprocessing strategy, we apply a windowing range of [-100, 200] to all involved CT scans for better contrast. Input images are firstly resized to $512 \times 512$ and then randomly cropped with a size of $448 \times 448$. To evaluate the generalization of MVG, we choose MSD Antonelli et al. (2022), a multi-organ segmentation dataset as an out-of-distribution dataset.

**Training details.** AdamW optimizer is used with a weight decay of 0.05. The peak learning is set to $1e^{-3}$ with a cosine learning rate scheduler. We train our model 100 epochs with 5 warm-up epochs.

| Method | AMOS CT | WORD | BTCV | AMOS MRI | MicroSegNet | PROMISE | Chest Defect | ACDC | LA |
|--------|---------|------|------|----------|-------------|---------|--------------|------|-----|
| | | | | *Specialists* | | | | | |
| ResNet-18 | 0.55 | 0.50 | 0.51 | 0.53 | 0.67 | 0.75 | 0.62 | 0.69 | 0.68 |
| UNet | 0.81 | 0.83 | 0.82 | 0.81 | 0.90 | 0.91 | 0.89 | 0.86 | 0.83 |
| VNet | 0.70 | 0.75 | 0.72 | 0.73 | 0.90 | 0.89 | 0.86 | 0.87 | 0.84 |
| TranUNet | 0.80 | 0.82 | 0.84 | 0.82 | 0.94 | 0.90 | 0.88 | 0.88 | 0.84 |
| nnUNet | 0.87 | 0.90 | 0.91 | 0.88 | 0.97 | 0.93 | 0.90 | 0.90 | 0.89 |
| | | | | *Generalists* | | | | | |
| UniverSeg* | 0.20 | 0.29 | 0.37 | 0.25 | 0.71 | 0.55 | 0.55 | 0.54 | 0.57 |
| Painter | 0.52 | 0.48 | 0.45 | 0.51 | 0.69 | 0.68 | 0.50 | 0.52 | 0.55 |
| LVM | 0.12 | 0.14 | 0.10 | 0.15 | 0.36 | 0.30 | 0.10 | 0.12 | 0.13 |
| SegGPT | 0.66 | 0.66 | 0.65 | 0.71 | 0.88 | 0.75 | 0.68 | 0.70 | 0.71 |
| MVG | 0.73 | 0.74 | 0.73 | 0.74 | 0.91 | 0.85 | 0.79 | 0.85 | 0.81 |

Table 2: **Quantitative evaluation in segmentation tasks.** Compared to other generalists, our method achieves state-of-the-art performance with solid improvements. *: We inference the official weights with 64 in-context samples from the training set.

We only adopt the random crop as the data augmentation. The sampling weight of segmentation tasks is 0.5 while the rest of the tasks share 0.5. We use 8 A5000 GPUs to train our models. We use 1 in-context sample for both training and inference.

**Training Objective.** In practice, for *all tasks except segmentation*, we perform 90% training iterations with *mask image modeling* and 10% training iterations with *auto-regressive* training. For *segmentation* tasks, we perform 100% training iterations with *auto-regressive* training.

**Evaluation.** We use mean IoU (mIoU) as the evaluation metric for segmentation. For cross-modal synthesis, inpainting, and denoising, we use mean absolute error (MAE), peak signal-to-noise ratio (PSNR), and structural similarity index measure (SSIM) as the evaluation metric.

## 4.2 A GENERALIST TO 13 MEDICAL TASKS

**Baselines.** Our generalist baselines includes LVM Bai et al. (2023) and Painter Wang et al. (2023a), which are trained on our benchmark. UniverSeg Butoi et al. (2023) is used as a segmentation generalist baseline. While our specialist baselines includes ResNet-18 He et al. (2016) with a two-layer MLP decoder, UNet Ronneberger et al. (2015), VNet Milletari et al. (2016), TransUNet Chen et al. (2021a), and nnUNet Isensee et al. (2021). For synthesis tasks, we involve Pix2Pix Isola et al. (2017) as an additional baseline.

**Quantitative evaluation.** In Table 2, we compare our method with the latest vision generalist modelsBai et al. (2023); Wang et al. (2023a) across a range of segmentation tasks. Our MVG achieves the best performance of 0.79 mIoU among all the generalists. Specifically, Our MVG outperforms Painter Wang et al. (2023a) by 0.24 mIoU, LVM Bai et al. (2023) by 0.62 mIoU on average, SegGPT by 0.09 mIoU on average, and UniverSeg Butoi et al. (2023) by 0.35 mIoU. All the generalists only require one model to perform these different tasks. UniverSeg is trained with up to 64 in-context samples, yet still yields inferior performance to our method which only relies on one in-context sample. We provide the visualization results in Figure 4 At the same time, specialist models like UNet Ronneberger et al. (2015), TranUNet Chen et al. (2021a), and nnUNet Isensee et al. (2021), which need to train different models for different tasks, still hold the edge in performance.

We report the image synthesis results in Table 3. we present a detailed quantitative comparison of our method with the latest generalist and specialist models across various tasks including cross-modal synthesis, inpainting, and denoising. Our MVG demonstrates strong capabilities and achieves competitive performance, particularly in the generalist category. For instance, in the task of cross-modal synthesis, MVG shows an improvement over Painter in all metrics: a lower MAE by 0.002, a higher PSNR by 0.69, and a better SSIM by 0.009 over the best vision generalist.

**Qualitative evaluation.** To provide a more intuitive observation of our MVG, we provide the visualization of different tasks in Figure 5.

**Other tasks.** We show that beyond segmentation, our MVG can handle more discriminative tasks such as object detection. Unlike standard object detection outputs, we form the output space as the

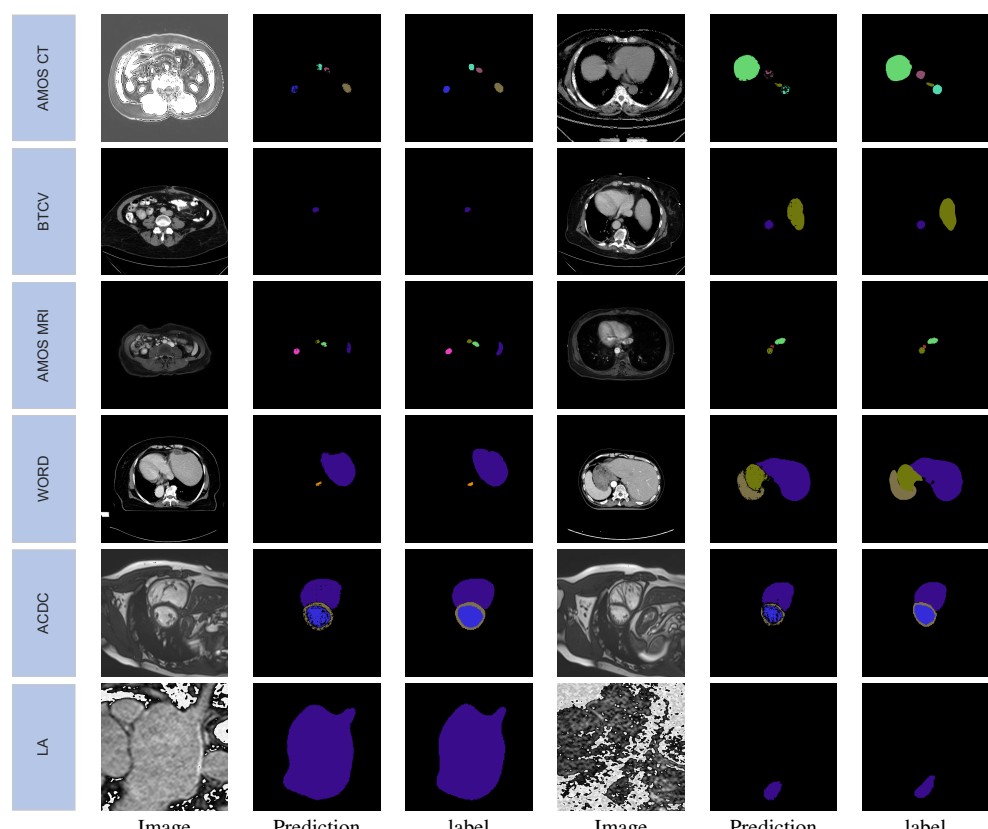

Figure 4: **Qualitative evaluation of segmentation.** MVG shows strong capabilities on various segmentation tasks covering multiple modalities and body regions.

| Method | Cross-modal synthesis | | | Inpaiting | | | Denoise | | |
|---|---|---|---|---|---|---|---|---|---|
| | MAE | PSNR | SSIM | MAE | PSNR | SSIM | MAE | PSNR | SSIM |
| *Specialists* | | | | | | | | | |
| ResNet-18 | 0.026 | 20.984 | 0.860 | 0.008 | 30.981 | 0.959 | 0.022 | 30.519 | 0.709 |
| Pix2Pix | 0.018 | 24.311 | 0.899 | 0.008 | 34.891 | 0.982 | 0.020 | 33.011 | 0.730 |
| TranUNet | 0.016 | 25.541 | 0.938 | 0.005 | 35.561 | 0.989 | 0.016 | 33.999 | 0.761 |
| *Generalists* | | | | | | | | | |
| Painter | 0.021 | 24.031 | 0.920 | 0.006 | 33.595 | 0.978 | 0.020 | 33.104 | 0.721 |
| MVG | 0.019 | 24.721 | 0.929 | 0.006 | 34.521 | 0.981 | 0.018 | 33.521 | 0.731 |

Table 3: **Quantitative comparison with other tasks.** Our model shows strong capabilities in the tasks of cross-modal synthesis, impainting, and denoising.

original image with the lesion's bounding box overlaid to indicate its location. Specifically, we also add a large-scale lesion detection dataset, DeepLesion Yan et al. (2018), when training MVG, which aims to identify and localize abnormalities in Chest CT images. These abnormalities include tumors, cysts, and other pathological changes within body tissues, organs, or bones. Our results, illustrated in Figure 7, demonstrate the efficacy of MVG in this context. This suggest that, in the future, we could train on images annotated with various types of labels—such as boxes, circles, and crosses—as provided by different human annotators. This would enable us to output image labels in the same format specified by prompts.

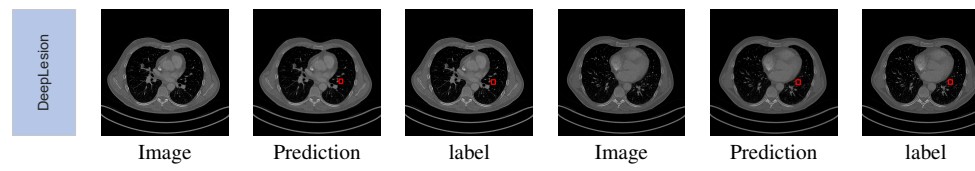

Figure 7: **Qualitative evaluation on DeepLesion detection dataset.**

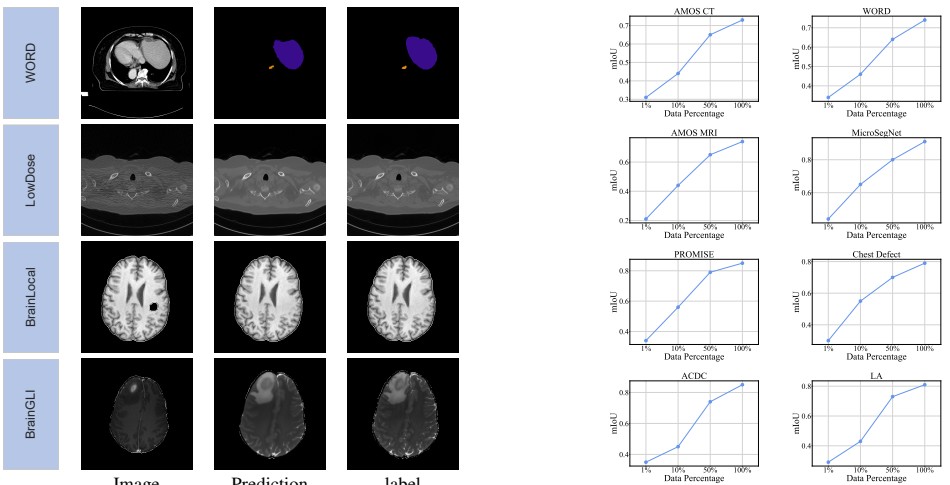

Figure 5: **Qualitative evaluation of four tasks:** Segmentation (1st row), denoising (2nd row), cross-modal synthesis (3th row), and inpainting (4th row).

Figure 6: **Impact of training data scale.** We ablate on various scales of the training data (randomly sampled from each dataset), ranging from 1% to 100%.

**Generalize to unseen datasets.** The advantage of in-context learning is that it allows models to adapt to new datasets quickly. We report the result in Table 4, MVG achieves 0.84 mIoU on MSD-Liver with only new prompts without any fine-tuning. After fine-tuning MVG with one instance on MSD-Spleen and MSD-Lung, MVG achieves 0.87 mIoU and 0.48 mIoU.

| Dataset | mIoU |
|---|---|
| MSD-Liver | 0.84 |
| MSD-Spleen | 0.87 |
| MSD-Lung | 0.48 |

Table 4: Generalization to the unseen MSD dataset.

### 4.3 ABLATION STUDY

**Data scalability** Nowadays, more and more datasets are available, which motivates us to study whether a scale-up dataset can train a stronger MVG. As shown in Table 6, we randomly choose 1%, 10%, 50% as comparison with the *full data*. The performance of our MVG consistently improves with the growth of dataset size. These results show the strong dataset scalability of our MVG.

**Color space** To unify the output space of different segmentation tasks in which the same value from different datasets may have different semantics, we propose to unify the output space with a pre-defined color for each class or random color that we keep the same semantics have the colors. As shown in Table 5, the random color performs much better than the pre-defined color on abdominal

| Method | AMOS CT | WORD | BTCV | AMOS MRI | MicroSegNet | PROMISE | Chest Defect | ACDC | LA |
|---|---|---|---|---|---|---|---|---|---|
| Binary | 0.46 | 0.48 | 0.48 | 0.49 | 0.78 | 0.78 | 0.45 | 0.49 | 0.52 |
| Pre-defined | 0.60 | 0.62 | 0.62 | 0.61 | 0.89 | 0.86 | 0.71 | 0.74 | 0.73 |
| Random | 0.73 | 0.74 | 0.73 | 0.74 | 0.91 | 0.85 | 0.79 | 0.85 | 0.81 |

Table 5: **Color space for segmentation.** Using semantic masks in a random color space as prompts significantly improves the segmentation performance of our generalist model.

| Method | AMOS CT | WORD | BTCV | AMOS MRI | MicroSegNet | PROMISE | Chest Defect | ACDC | LA |
|--------|---------|------|------|----------|-------------|---------|--------------|------|-----|
| Isolated | 0.55 | 0.57 | 0.57 | 0.58 | 0.80 | 0.77 | 0.70 | 0.69 | 0.68 |
| Unified | 0.73 | 0.74 | 0.73 | 0.74 | 0.91 | 0.85 | 0.79 | 0.85 | 0.81 |

Table 6: **Isolated vs. Unified training.** "Isolated" indicates training our MVG individually on each dataset, while "Unified" indicates training on all datasets together.

| Method | AMOS CT | WORD | BTCV | AMOS MRI | MicroSegNet | PROMISE | Chest Defect | ACDC | LA |
|--------|---------|------|------|----------|-------------|---------|--------------|------|-----|
| MIM (mask 50%) | 0.56 | 0.48 | 0.46 | 0.54 | 0.70 | 0.66 | 0.50 | 0.50 | 0.52 |
| MIM (mask 75%) | 0.53 | 0.42 | 0.44 | 0.52 | 0.70 | 0.63 | 0.48 | 0.50 | 0.51 |
| Auto-regressive | 0.73 | 0.74 | 0.73 | 0.74 | 0.91 | 0.85 | 0.79 | 0.85 | 0.81 |

Table 7: **Auto-regressive training boosts in-context segmentation.** Randomly masking can harm in-context segmentation task, especially when it results in the complete removal of small organs. Auto-regressive training addresses this weakness and makes much better performance than MIM.

segmentation while having a similar performance on prostate segmentation. In particular, our MVG with random color space gains the average result of 0.735 mIoU on abdominal segmentation and improves 0.123 mIoU over that with pre-defined color space. In contrast, our MVG achieves inferior performance with pre-defined or random colors. The random color makes MVGs learn more from the context instead of the color itself and avoid the model being limited by the number of colors.

**Isolated and Unified training** To validate that our MVG can benefit from large-scale datasets across different tasks. We compare two settings: 1) isolated training: we train different models on different datasets in isolation. Namely, we train 13 models for the 13 datasets. 2) unified training: we train our MVG on all datasets together. Note that both settings have the same model architecture. We report the results in Table 6. The unified model makes significant improvements over the isolated model in all tasks and the improvements reach 0.14 mIoU. Such results indicate that MVG can benefit from large-scale datasets even if this dataset has different annotation semantics which motivates the medical image analysis community to further expand the datasets.

**Auto-regressive v.s. MIM** We ablate the conditional image generation methods: mask image modeling and auto-regressive training. As shown in Table 7, Auto-regressive training emerges as the significantly superior method, outperforming MIM across all datasets. The results underscore the fundamental weakness of random masking strategies in segmentation tasks, especially when dealing with small anatomical organs. By leveraging medical images' inherent spatial and contextual information, auto-regressive training offers a powerful alternative that significantly enhances in-context learning and segmentation performance.

## 5 CONCLUSION

In this work, we present MVG, a versatile model capable of handling various medical imaging tasks, including cross-modal synthesis, segmentation, denoising, and inpainting, within a unified image-to-image generation framework. MVG employs an in-context generation strategy to standardize inputs and outputs as images, allowing flexible task unification across various modalities and datasets. A hybrid approach combining masked image modeling and autoregressive training proved the most effective. To thoroughly assess MVG's potential and limitations, we also curate the first comprehensive generalist medical vision benchmark consisting of 13 datasets across 4 imaging modalities, including CT, MRI, X-ray, and Micro-ultrasound. Experiment results demonstrate that MVG consistently outperforms existing vision generalists. Benefiting from the in-context learning scheme, MVG demonstrates exceptional flexibility, scalability, and potential for generalization to unseen datasets with minimal samples. We will make our code and benchmark publicly available to encourage future research in medical AI generalists. We believe MVG can serve as a stepstone to make medical imaging tools more accessible to clinical researchers, other scientists, or beyond, by lowering the bar of machine learning expertise, computational resources, and human labor.

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
