# OpenReview forum: "Medical Vision Generalist: Unifying Medical Imaging Tasks in Context"
_ICLR.cc/2025/Conference — ICLR 2025 Conference Withdrawn Submission_

### Official Review · Reviewer_vfk8 · 2024-10-29

**Soundness:** 3
**Presentation:** 4
**Contribution:** 3
**Rating:** 6
**Confidence:** 5

**Summary:**

In this paper, the authors introduce a unified framework, named Medical Vision Generalist, for medical imaging analysis tasks, including segmentation, cross-modality synthesis, inpainting and denoising. The authors formulate the learning task as prompt-based learning task and prompt include task-image and task-label pairs. The authors adopt a single-channel colorization method to unify the output space of images across tasks. The authors combine the masked imaging modeling and auto-aggressive training method to train the model. To demonstrate the framework's capabilities, the authors curate a comprehensive generalist medical vision benchmark.

**Strengths:**

- The paper is well written and easy to follow.
- The paper proposes a solution to address the generalization of medical imaging analysis, such as cross-domain problem.
- The paper proposes a unified colorization formulation to unify the different output types of medical imaging analysis tasks.
- The paper treats the different learning tasks as a prompt-based learning task.
- The paper introduces a new benchmark for generalist medical imaging analysis.

**Weaknesses:**

- The proposed framework is limited to 2D scenario.
- The paper does not explain the clinical motivation why a generalized medical imaging analysis model is needed. To me, since there are many pre-trained specialized models, medical researchers can just pick one of the state-of-the-art models and get better performance than the generalized models.
- The motivation of combining the masked imaging modeling and auto-aggressive is not clear. In the experiment, auto-aggressive training is more superior than masked imaging modeling. Then why combining them in the first place?
- The generalization of proposed framework is not very convincing, since the organs in the unseen-dataset also appears in training set. What about other types of medical imaging dataset, such as pathology data?

**Questions:**

- Could the authors explain the potential clinical benefits of your proposed framework in detail?
- Could the authors add more unseen-dataset evaluation to test the generalization of the proposed framework, for example cell data whose semantic is not available in the training data?
- Could the authors explain architecture of your ViT, such as the embedding size or number of layers or the total number of parameters, since the network requires 8 A5000 gpus to train?
- Could the authors explain the training time of auto-aggression modeling?
- Could the authors explain the inference resources needed of your model? Can it run on CPU under an acceptable time? How much time does it require to infer a given test image under a given test hardware?

---

> ### Author Response · Authors · 2024-11-27
> **Rebuttal by authors -- Part I**
>
> Thanks for the appreciation of this work. We address your concerns below:
>
> Q1: The proposed framework is limited to 2D scenario.
>
> A1: ​​Thank you for pointing out this important concern. Our current framework operates on 2D images, similar to approaches like UniverSeg. While we acknowledge the critical importance of 3D contextual information for accurately analyzing anatomical structures and the limitations of 2D analysis in capturing volumetric relationships, transitioning directly to 3D models presents significant computational challenges that require careful, non-trivial design. How to properly incorporate 2.5D or fully 3D models into MVG will be thoroughly investigated as future study. We will revise the manuscript to explicitly clarify this limitation and provide a roadmap for integrating 3D capabilities.
>
>
> Q2: clinical motivation. To me, since there are many pre-trained specialized models, medical researchers can just pick one of the state-of-the-art models and get better performance than the generalized models.
>
> A2: For domains with well-established, large-scale pre-trained models, selecting one of the state-of-the-art specialized models for optimal performance is feasible. However, for domains lacking such pre-trained models, we argue that generalist models, trained on far more diverse datasets, hold a significant advantage in transfer learning.
> Generalist models offer unique benefits by consolidating diverse functionalities and insights within a unified framework. This integration enables them to address multiple tasks or domains within a single system, eliminating the need to develop and train numerous specialized models. This minimizes development time and provides a more cost-effective solution for researchers, particularly in resource-constrained settings.
> From a performance perspective, while generalist models underperform compared to specialized models across benchmarks for both natural and medical images, their robust scalability is a promising indicator of future potential. As the availability of data and computational resources increases, the generalist approach may evolve to surpass specialized models in performance, offering a comprehensive solution that balances versatility and efficiency.
>
> Q3:  The motivation of combining the masked imaging modeling and auto-aggressive is not clear. In the experiment, auto-aggressive training is more superior than masked imaging modeling. Then why combining them in the first place?
>
> A3: Sorry for the confusion. While AR training outperforms MIM for image segmentation, this superiority does not extend to other tasks.
> MIM’s suboptimal performance in segmentation, as shown in Table 7, arises from its masking strategy, which can disrupt the preservation of global contextual information crucial for delineating anatomical structures, such as the spatial relationships among abdominal organs. This aligns with findings from [1], which suggest that MIM is better suited for capturing local details but struggles with maintaining global context.
> However, for tasks like inpainting and denoising, where refining local details takes precedence over preserving global context, MIM can be quite beneficial. To validate this, we compared MIM and AR training for cross-modal synthesis, inpainting, and denoising tasks using mean absolute distance as the evaluation metric, as shown in the Table below. MIM consistently outperformed AR training in these tasks while also requiring fewer computational resources.
> Based on these insights and the results in Table 7, we adopted a hybrid training strategy:
> For segmentation tasks, AR training is used exclusively.
> For all other tasks, we allocate 90% of training iterations to MIM and 10% to AR training to leverage the strengths of both methods.
> | Method          | GPU Hours (RTX A5000) | Cross-modal Synthesis | Inpainting | Denoise | MIM   |
> |------------------|-----------------------|------------------------|------------|---------|-------|
> | Mask Image Modeling (MIM) | 972h                 | 0.019                  | 0.006      | 0.018   |       |
> | Autoregressive   | 1062h                | 0.020                  | 0.006      | 0.019   |       |

---

> ### Author Response · Authors · 2024-11-27
> **Rebuttal by authors -- Part Ⅱ**
>
> Q4: What about other types of medical imaging datasets, such as pathology data?
>
> A4: Pathology images, which are typically 3-channel RGB images, present significant differences from the radiology images (e.g., CT and MRI) used in our current training data due to domain and modality gaps. These gaps arise from the fact that pathology images are collected and scanned in entirely different environments. As a result, our MVG cannot be directly applied to pathology datasets in its current form without pretraining on such data.
> We plan to enhance the diversity of data types in the pretraining stage by incorporating pathology images and other imaging modalities in future work. This can further enhance the MVG’s generalization capabilities across a broader range of medical imaging datasets, improving its robustness and applicability.
>
> Q5: Could the authors explain the architecture of your ViT, since the network requires 8 A5000 gpus to train?
>
> A5: We use ViT large with depth=24 and width=1024. To finish training on A5000, we use gradient checkpoint and accumulation to reduce memory consumption.
>
> Q6: training time.
>
> A6: The total training time is 1,062 GPU (A5000) hours
>
> Q7: inference resources
>
> A7: Our model is optimized for GPU deployment, taking full advantage of the parallel processing capabilities of modern GPUs to achieve high inference efficiency. On an RTX A5000 GPU, our model processes a single test image in approximately 0.3 seconds. While the model can run on a CPU, the inference speed is significantly slower (~3.7 seconds), making GPU utilization essential for maintaining acceptable processing times in demanding environments. To further improve performance for CPU deployment, we plan to explore advanced model compression techniques [2,3] and acceleration methods [2].
>
>
> [1] Xie, Z., Geng, Z., Hu, J., Zhang, Z., Hu, H., & Cao, Y. (2023). Revealing the dark secrets of masked image modeling. In Proceedings of the IEEE/CVF conference on computer vision and pattern recognition (pp. 14475-14485).
>
> [2] Choudhary, T., Mishra, V., Goswami, A., & Sarangapani, J. (2020). A comprehensive survey on model compression and acceleration. Artificial Intelligence Review, 53, 5113-5155.
>
> [3] Li, Z., Li, H., & Meng, L. (2023). Model compression for deep neural networks: A survey. Computers, 12(3), 60.

---

> > ### Author Response · Authors · 2024-11-30
> >
> > Dear Reviewer vfk8,
> >
> > We sincerely appreciate your review. We have carefully considered each of your questions and provided detailed responses in the rebuttal. Please let us know if you have any further questions or concerns.
> >
> > Thanks!

---

> > > ### Author Response · Authors · 2024-12-01
> > >
> > > Dear Reviewer vfk8,
> > >
> > > We would like to kindly remind you that the discussion deadline is approaching. After this deadline, we may not have the opportunity to respond.
> > >
> > > We sincerely appreciate your time and thoughtful input and look forward to hearing your feedback.
> > >
> > > Sincerely,
> > >
> > > Authors of Submission 8515

---

### Official Review · Reviewer_crRU · 2024-11-03

**Soundness:** 2
**Presentation:** 2
**Contribution:** 2
**Rating:** 5
**Confidence:** 4

**Summary:**

The paper introduces a medical vision generalist (MVG) model, which unifies various medical imaging tasks including segmentation, cross-modal synthesis, denoising, and inpainting within a single image-to-image generation framework. MVG utilizes in-context learning, treating tasks as image generation processes conditioned on prompt image-label pairs, allowing for flexible adaptation to different modalities and datasets. The authors have also curated a comprehensive benchmark with 13 datasets across four imaging modalities to evaluate MVG, which consistently outperforms existing generalist models.

**Strengths:**

The paper first uses in-context learning to unify multiple medical vision tasks, which is original.
The proposed output space unification strategy is useful when training with multiple segmentation tasks.

**Weaknesses:**

The advantage of unifying multiple medical vision tasks through the MVG model could not be verified based on the evidence provided in this paper.
According to Table 2 and Table 5, the performance gain in segmentation tasks could be due to the colorization strategy instead of unifying other vision tasks.
Regarding cross-modal synthesis, inpainting, and Denoising tasks, the improvement by the MVG model is marginal compared to the previous generalist model and all generalist models performs worse than specialist models in each task according to Table 3.
Besides, the experiments on the scalability were conducted on several small datasets, which could not demonstrate the potential to increase the performance when unifying all datasets.

**Questions:**

To better demonstrate the advantage of unifying multiple medical vision tasks through the MVG model, the authors could make the following efforts to address the concerns,
1. compare the segmentation performance while isolating the effects of the colorization strategy from task unification.
2. discuss potential reasons for the marginal improvements in these tasks and propose strategies to close the gap with specialist models
3. conduct scalability experiments on the unified datasets.

---

> ### Author Response · Authors · 2024-11-27
> **Rebuttal by authors**
>
> Thanks for your comments. We address your concerns below:
>
> Q1: compare the segmentation performance while isolating the effects of the colorization strategy from task unification.
>
> A1: We clarify that colorization is key to task unification in our method, as it inherently unifies all tasks. In Table 5, we conduct an ablation study to isolate the effects of different colorization strategies while maintaining task unification, allowing us to evaluate colorization's impact independently. Specifically, MVG attains average IoU scores of 0.55, 0.71, and 0.79 across nine segmentation datasets using binary, pre-defined, and random colorization, respectively.
>
> Q2: Regarding cross-modal synthesis… the improvement by the MVG model is marginal compared to the previous generalist model, and all generalist models perform worse than specialist models in each task, according to Table 3.
>
> A2:  The improvement achieved by MVG is not marginal. As shown in Table 3, our method consistently outperforms in all three low-level tasks (synthesis, inpainting, and denoising) across all three evaluation metrics (MAE, SSIM, and PSNR). In addition to the results presented in Figure 5, we will include further qualitative comparisons with existing generalist models to better highlight MVG's effectiveness.
> Our primary focus is on demonstrating how our pipeline enables the integration of multiple medical vision tasks into a unified image-to-image generation framework, representing a novel vision-centric approach to medical generalist models. While generalist models currently underperform specialized models, they offer greater flexibility. Unlike specialized models like U-Net or nnUNet, which are limited to segmentation tasks, MVG can be trained across a wide range of tasks and easily adapted to new datasets with minimal task-specific data.
> Moreover, MVG’s strong scalability—both in terms of data and task diversity—suggests that, with increased data and computational resources, our generalist approach has the potential to surpass specialized models. Unlike specialized models, which are limited to a single task and cannot benefit from task-level scalability, MVG's ability to handle a wide range of tasks positions it to scale more effectively as new tasks and data emerge. This inherent flexibility gives our generalist model advantages in scaling to meet the evolving demands of medical imaging.
>
> Q3: the experiments on the scalability were conducted on several small datasets, which could not demonstrate the potential to increase the performance when unifying all datasets.
>
> A3: We first assemble all 12 training datasets into a large training dataset including Segmentation: 212,811 pairs, Synthesis: 1,987,111 pairs, Inpainting: 139,311 pairs, Detection: 66,532 pairs. All scalability experiments were conducted on this comprehensive dataset, which includes over 2 million pairs. Metrics for each sub-task were reported based on this unified dataset, demonstrating the framework's ability to scale effectively across diverse tasks. Additionally, we plan to expand these experiments to incorporate even more datasets in future work to further validate the scalability of our approach.

---

> > ### Author Response · Authors · 2024-11-30
> >
> > Dear Reviewer crRU,
> >
> > We sincerely appreciate your review. We have carefully considered each of your questions and provided detailed responses in the rebuttal. Please let us know if you have any further questions or concerns.
> >
> > Thanks!

---

> > > ### Author Response · Authors · 2024-12-01
> > >
> > > Dear Reviewer crRU,
> > >
> > > We would like to kindly remind you that the discussion deadline is approaching. After this deadline, we may not have the opportunity to respond.
> > >
> > > We sincerely appreciate your time and thoughtful input and look forward to hearing your feedback.
> > >
> > > Sincerely,
> > >
> > > Authors of Submission 8515

---

> > ### Comment · Reviewer_crRU · 2024-12-02
> > **Response to the rebuttal**
> >
> > Thanks for the detailed response by the authors
> >
> > The comments have clarified the concerns about the colorization strategy(Q1) and scalability (Q3).
> >
> > Regarding Q2, the authors claimed that the primary focus is on demonstrating how the proposed pipeline enables the integration of multiple medical vision tasks. However, current experiments could not provide enough evidence since the comparisons (Table 6) are performed between training with unified and isolated datasets. It is recommended that the authors perform ablation study on vision tasks instead of datasets. Besides, the practical applicability of the proposed method is limited, given that its performance is lower than specialized models in almost all tasks.
> >
> > Therefore, the manuscript is below the acceptance threshold.

---

### Official Review · Reviewer_UcLG · 2024-11-04

**Soundness:** 2
**Presentation:** 3
**Contribution:** 2
**Rating:** 3
**Confidence:** 4

**Summary:**

This paper introduces Medical Vision Generalist (MVG), a foundational model designed to handle diverse medical imaging tasks within a unified framework. MVG standardizes input and output as images through an in-context generation approach, treating tasks as an image generation process conditioned on prompt image-label pairs. To leverage both local and global context, MVG combines masked image modeling with autoregressive training for conditional image generation. Experimental results indicate that MVG outperforms existing vision generalists, such as Painter and LVM, demonstrating scalability and adaptability across modalities and datasets.

**Strengths:**

1. The authors insightfully recognize that a pure masking strategy is insufficient for medical image segmentation, leading them to incorporate an autoregressive training pipeline. Experimental results confirm the effectiveness of this approach.

2. Unlike other foundational models focused primarily on segmentation, this paper addresses a broader range of tasks, including cross-modal synthesis, image denoising, and inpainting, opening potential new research directions.

**Weaknesses:**

1. While the addition of multiple tasks is beneficial, the paper overlooks essential medical imaging tasks, such as image registration and inverse reconstruction, making MVG appear more like an expanded segmentation model than a comprehensive foundation model. The reviewer suggests that MVG’s learned feature representations could potentially support image registration by integrating a flow estimation head, and inverse reconstruction by using denoising as a regularizer, unfolding the inverse optimization problem with forward consistency [1] within the network.

2. Restricting training to 2D images raises concerns about MVG’s utility as a foundational model for medical imaging. Effective 3D analysis is crucial, as many anatomical structures (e.g., brain cortex, lung vessels, heart) span significant volumes, where 2D slices may miss critical contextual information.

3. The authors claim that MVG “scales well with multiple tasks and datasets,” yet the evidence provided in Figure 6 and Table 6 only demonstrates that more data improves performance, a known property of deep learning models, and that unified training is preferable to isolated training, which reiterates established insights for vision transformers requiring large-scale data.

4. The paper lacks runtime and complexity analysis, particularly GPU resources for training. Comparisons with resource-efficient models, such as nnU-Net trained on individual datasets, would offer a clearer picture. Specifically, what GPU/hours are required for training MVG on all datasets, and how does GPU memory usage compare to nnU-Net or other specialist models trained on individual datasets?
[1] MoDL: Model-Based Deep Learning Architecture for Inverse Problems, TMI 2019.

**Questions:**

1. Can the authors explain the discrepancy in ACDC Dice scores for UniverSeg between Table 2 in this paper (0.54) and Table 6 in the UniverSeg paper appendix (0.70)?

2. Is the failure of mask modeling in segmentation tasks due to the use of L1 loss? Have alternative segmentation losses, such as Dice loss, been tested?

3. Could the authors elaborate on the motivation and practical benefits of MVG in medical imaging? Clinically, how does a foundational model outperform specialist models in practice, such as in speed, ease of deployment, or cost-effectiveness? Technically, what key insights from MVG could guide future researchers in developing clinically and economically viable foundational models?

---

> ### Author Response · Authors · 2024-11-27
> **Rebuttal by authors -- Part I**
>
> Thanks for your helpful comments. We address your concerns below:
>
> Q1: More medical imaging tasks. While the addition of multiple tasks is beneficial, the paper overlooks essential medical imaging tasks, such as image registration and inverse reconstruction, making MVG appear more like an expanded segmentation model than a comprehensive foundation model. The reviewer suggests that MVG’s learned feature representations could potentially support image registration by integrating a flow estimation head and inverse reconstruction by using denoising as a regularizer, unfolding the inverse optimization problem with forward consistency [1] within the network.
>
> A1: We respectfully disagree with the characterization of MVG as an expanded segmentation model. While segmentation datasets in the benchmark are more numerous compared to other tasks, the total size of data used for low-level tasks (e.g., denoising and inpainting) is substantially larger. Specifically, the benchmark includes 2,475,636 training image/label pairs, distributed as follows:
> **Segmentation - 212,811 pairs**,
> **Synthesis - 1,987,111 pairs**,
> **Inpainting - 139,311 pairs**,
> **Detection - 66,532 pairs**,
> **Denoising - 69,871 pairs**.
>
> As mentioned in line 340 of the original paper, the sampling weight of segmentation tasks is 0.5 while the rest of the tasks share 0.5. Therefore, all tasks are well balanced and MVG is not dominated by a certain task.
> MVG’s key contribution lies in unifying diverse medical imaging tasks within a single image-to-image generation framework, rather than employing task-specific fine-tuning or additional task-specific heads. However, our framework can be indeed extended to other tasks with modifications. For instance, adding a flow estimation head enables image registration. Following [2], we trained and evaluated MVG on the Neurite-OASIS dataset [3], consisting of 350 training slices and 64 testing slices, achieving a Dice score of 0.602:
> |Method       | Registration |
> |--------------|--------------|
> |       MVG        | 0.602        |
>
> For inverse reconstruction, the task requires 12-frame inputs, which would significantly increase computational demands and training costs—an addition that is non-trivial given the current computation constraints. This challenge is compounded by the already high training budget (see Q4 below), driven by the large input size (4× the original image size, see Figure 3) and the use of the ViT-Large architecture.
> We acknowledge the importance of these tasks and will discuss their applicability in the manuscript, referencing [1]. In future work, we will explore efficient approaches to extend MVG for multi-frame input tasks and other applications.
>
> Q2: Restricting training to 2D images raises concerns about MVG’s utility as a foundational model for medical imaging. Effective 3D analysis is crucial, as many anatomical structures (e.g., brain cortex, lung vessels, heart) span significant volumes, where 2D slices may miss critical contextual information.
>
> A2: ​​Our current framework operates on 2D images, similar to approaches like UniverSeg. While we acknowledge the critical importance of 3D contextual information for accurately analyzing anatomical structures and the limitations of 2D analysis in capturing volumetric relationships, transitioning directly to 3D models presents significant computational challenges that require careful, non-trivial design. How to properly incorporate 2.5D or fully 3D models into MVG will be thoroughly investigated as future study. We will revise the manuscript to explicitly acknowledge this limitation and outline a roadmap for integrating 3D capabilities into MVG.

---

> > ### Comment · Reviewer_UcLG · 2024-12-02
> >
> > Many thanks to the authors for their detailed response to the concerns raised by the reviewer.
> >
> > **A1**
> > The reviewer characterizes MVG as an expanded segmentation model not because it incorporates additional tasks into a universal segmentation model, but due to the lack of clarity in its motivation and the specific problem it aims to address. From a clinical perspective, the additional tasks, such as cross-modal synthesis and inpainting, have not yet demonstrated clear advantages in real-world clinical practice. This makes these tasks appear incremental relative to the core task of image segmentation. Moreover, the reported registration performance of 60% Dice (a marginal improvement over the initial alignment) suggests a fundamental misunderstanding of what constitutes a clinically useful tool. Tools intended for clinical applications require both robustness and significant improvements to justify their utility in practice, which is beyond what current form of MVG can provide.
> >
> > **A3**
> > While the authors fail to demonstrate the clinical effectiveness of MVG, as outlined in **A1**, it is also important to evaluate the contributions from a technical perspective. Technically, the study claims to demonstrate, for the first time, that heterogeneous tasks can be coherently trained together within a unified and scalable framework for medical imaging. However, the MVG primarily borrows methodologies from the computer vision community. While leveraging advances from other domains is valid, the lack of demonstrated clinical relevance, as discussed in **A1**, further undermines its overall impact.
> >
> > Additionally, the authors’ statement in **A6**, that "the limitations of masked image modeling (MIM) in segmentation tasks stem from its focus on recovering local details, which compromises global contextual information crucial for segmentation," suggests a misunderstanding of many state-of-the-art masked image modeling techniques. For instance, IJEPA [1] effectively uses masked image modeling in the latent space, addressing precisely the concerns raised by the authors.
> >
> > The paper, in its current form, is far from being ready for publication at ICLR. The reviewer suggests that the authors focus on refining the manuscript to emphasize either advancing clinical translation or pushing the technical boundaries. A clearer direction in one of these areas would strengthen the paper’s contribution and relevance.
> >
> > [1] Yann LeCun et al. IJEPA: Implicit Joint Embedding Predictive Architectures. ICCV 2023.

---

> ### Author Response · Authors · 2024-11-27
> **Rebuttal by authors -- Part Ⅱ**
>
> Q3: The authors claim that MVG “scales well with multiple tasks and datasets,” yet the evidence provided in Figure 6 and Table 6 only demonstrates that more data improves performance, a known property of deep learning models and that unified training is preferable to isolated training, which reiterates established insights for vision transformers requiring large-scale data.
>
> A3: Thank you for raising this important point. However, for deep networks, data scaling does not always work well. For example, CNNs face more significant scaling challenges compared to Transformers [5]. Moreover, even within Transformers, those trained using supervised learning exhibit poorer scaling capabilities than their self-supervised counterparts [6,7,8]. These findings emphasize the need to design effective algorithms with robust scaling properties and to rigorously evaluate their scaling performance, as demonstrated in our work.
>
> To our knowledge, this is the first study to propose and implement a unified learning framework for training a generalist model across diverse medical vision tasks, spanning both high-level objectives (e.g., segmentation) and low-level ones (e.g., denoising, inpainting, and synthesis). Unlike prior work, which typically focuses on scalability within isolated tasks or domains (e.g., MedSAM and UniverSeg for segmentation), **our study demonstrates for the first time that heterogeneous tasks can be coherently trained together in a unified and scalable framework**.
> MVG validates the framework’s adaptability to emerging datasets and tasks, providing a versatile solution for the evolving needs of medical imaging. 1) While the feasibility of a generalist medical AI that unifies multiple imaging tasks has remained unclear, our study offers the first proof-of-concept for a vision-centric generalist model capable of handling diverse tasks within a unified image-to-image generation framework. 2) We also show that MVG exhibits strong scalability, improving performance with more diverse training tasks and effectively adapting to new datasets with minimal task-specific samples. This highlights the potential of generalist models to address future medical imaging challenges.
> 3) To advance understanding, we provide an in-depth analysis of the roles of masked image modeling and autoregressive training in developing a generalist medical model, an aspect not explored in prior studies. Our analysis also includes ablation studies on objective functions, data/task balancing, and generalization to new discriminative tasks, such as detection.
> These contributions go far beyond reiterating known properties, offering novel insights into the feasibility and potential of unified training across diverse medical domains.
>
> Q4: runtime and complexity analysis.
> A4: Thanks for the suggestion. For a fair comparison, we report the training cost of TransUnet and our MVG evaluated on 1 A5000 GPU.
> | Method    | Param | Training Cost |
> |-----------|-------|---------------|
> | MVG       | 370M  | 1062h         |
> | TransUnet | 92M   | 250h          |
>
> Note that a generalist usually costs more training time than a specialist. MVG requires more training resources due to 1) the significantly larger input dimensions (our framework uses inputs that are four times larger, including the prompt image/label and task image) during training and 2) the use of a larger network (ViT-Large) compared to standard segmentation methods, such as UNet.
>
> Q5: the discrepancy in ACDC Dice scores for UniverSeg between Table 2 in this paper (0.54) and Table 6 in the UniverSeg paper appendix (0.70)?
>
> A5: The discrepancy arises from differences in the testing procedure. In the UniverSeg paper, only the max (slice with the maximum area of the target mask) and mid slices were tested. In our work, we randomly sampled from all slices to create a more comprehensive testing set, resulting in a different testing split.
>
>
> Q6: Is the failure of mask modeling in segmentation tasks due to the use of L1 loss? Have alternative segmentation losses, such as Dice loss, been tested?
>
> A6: Thank you for the insightful question. As noted in Lines 254–263 of the original manuscript, the limitations of masked image modeling (MIM) in segmentation tasks stem from its focus on recovering local details, which compromises global contextual information crucial for segmentation. This aligns with findings from [4], which show that MIM is better suited for capturing local rather than global information. Since segmentation relies heavily on global context to delineate anatomical structures, this issue should persist regardless of the choice of loss function. To further validate this point, we conducted experiments on the AMOS CT dataset and compared Dice loss and L1 loss. The results indicate similar performance (Dice: 0.48 vs. L1: 0.46).
>
> | L1 Loss    | 0.46  |
> |------------|-------|
> | Dice Loss  | 0.48  |

---

> ### Author Response · Authors · 2024-11-27
> **Rebuttal by authors -- Part Ⅲ**
>
> Q7: Clinical practice. Could the authors elaborate on the motivation and practical benefits of MVG in medical imaging?
>
> A7: Generalist models typically require more computational resources because they need to handle a wide range of tasks simultaneously. Nevertheless, generalist models still have non-negligible advantages over specialized models. They provide versatility by handling multiple tasks or domains within a single framework, **reducing the need for separate models and minimizing development time**. Their ability to leverage **shared knowledge across tasks improves data efficiency and facilitates scalability as new tasks emerge**. Additionally, generalist models often excel in transfer learning, offering cost-effective solutions by integrating various functionalities and insights into one unified system.
>
> In this research work, our primary focus is on demonstrating how our pipeline enables the integration of multiple medical vision tasks into a unified image-to-image generation framework, representing a novel vision-centric approach to medical generalist models. While generalist models currently underperform specialized models across various benchmarks—spanning both natural and medical imaging domains—they offer transformative potential by **unifying multiple modalities, tasks, and datasets into a single framework**. This new paradigm provides **far greater flexibility** compared to task-specific models—unlike specialized models which are limited to segmentation tasks, MVG can be directly trained across a wide range of tasks and easily adapted to new datasets with minimal task-specific data.
>
> Moreover, MVG’s strong **scalability—both in terms of data and task diversity**—suggests that, with increased data and computational resources, our generalist approach has the potential to surpass specialized models. Unlike specialized models, which are limited to a single task and cannot benefit from task-level scalability, MVG's ability to handle a wide range of tasks positions it to scale more effectively as new tasks and data emerge. In addition to scalability, this flexibility also promotes task generalization and efficient knowledge transfer, making MVG a forward-compatible solution to meet the evolving demands of medical imaging. With increased data and computational resources, our generalist approach may have the potential to surpass specialized models in the future to meet the evolving demands of medical imaging.
>
> [1] Aggarwal, H. K., Mani, M. P., & Jacob, M. (2018). MoDL: Model-based deep learning architecture for inverse problems. IEEE transactions on medical imaging, 38(2), 394-405.
>
> [2] Balakrishnan, G., Zhao, A., Sabuncu, M. R., Guttag, J., & Dalca, A. V. (2019). Voxelmorph: a learning framework for deformable medical image registration. IEEE transactions on medical imaging, 38(8), 1788-1800.
>
> [3] Hoopes, A., Hoffmann, M., Greve, D. N., Fischl, B., Guttag, J., & Dalca, A. V. (2022). Learning the effect of registration hyperparameters with hypermorph. The journal of machine learning for biomedical imaging, 1.
>
> [4] Xie, Z., Geng, Z., Hu, J., Zhang, Z., Hu, H., & Cao, Y. (2023). Revealing the dark secrets of masked image modeling. In Proceedings of the IEEE/CVF conference on computer vision and pattern recognition (pp. 14475-14485).
>
> [5] Dosovitskiy, A. (2020). An image is worth 16x16 words: Transformers for image recognition at scale. arXiv preprint arXiv:2010.11929.
>
> [6] Radford, A. (2018). Improving language understanding by generative pre-training.
>
> [7] Kenton, J. D. M. W. C., & Toutanova, L. K. (2019, June). Bert: Pre-training of deep bidirectional transformers for language understanding. In Proceedings of naacL-HLT (Vol. 1, p. 2).
>
> [8] He, K., Chen, X., Xie, S., Li, Y., Dollár, P., & Girshick, R. (2022). Masked autoencoders are scalable vision learners. In Proceedings of the IEEE/CVF conference on computer vision and pattern recognition (pp. 16000-16009).

---

> > ### Author Response · Authors · 2024-11-30
> >
> > Dear Reviewer UcLG,
> >
> > We sincerely appreciate your review. We have carefully considered each of your questions and provided detailed responses in the rebuttal. Please let us know if you have any further questions or concerns.
> >
> > Thanks!

---

> ### Author Response · Authors · 2024-12-01
>
> Dear Reviewer UcLG,
>
> We would like to kindly remind you that the discussion deadline is approaching. After this deadline, we may not have the opportunity to respond.
>
> We sincerely appreciate your time and thoughtful input and look forward to hearing your feedback.
>
> Sincerely,
>
> Authors of Submission 8515

---

### Official Review · Reviewer_Q4Bs · 2024-11-05

**Soundness:** 4
**Presentation:** 3
**Contribution:** 4
**Rating:** 6
**Confidence:** 5

**Summary:**

The work is about a new foundation model for medical image analysis that aims to unify multiple imaging tasks (segmentation, cross-modal synthesis, inpainting, and denoising), called Medical Vision Generalist (MVG). The implementation is within a single model using a standardized image-to-image generation framework. MVG distinguishes itself by employing in-context learning strategies, which eliminate the need for retraining on new datasets and enable quick adaptation to unseen tasks with minimal labeled samples. Also authors introduce a benchmark that covers 13 datasets across 4 medical tasks for various imaging modalities (CT, MRI, X-ray, and micro-ultrasound). The latter makes this study not only about the methodology but also an important contribution to the research community.

**Strengths:**

The comprehensiveness of imaging modalities and tasks. The paper addresses segmentation, cross-modal synthesis, inpainting, and denoising tasks across various modalities, like CT, MRI, X-ray, and micro-ultrasound.

High performance and generalization ability is remarkable. MVG outperforms SOTA models such as Painter and LVM in most metrics. It also demonstrates scalability with different datasets and adaptability to unseen datasets with minimal samples.

Methodology of training - hybrid approach with masked image modeling and autoregressive training is a novelty that provides strong performance across diverse tasks. Also this approach partly solves the problem with medical imaging data augmentation (like cropping operations).

**Weaknesses:**

As shown by authors, the hybrid use of autoregressive training boosts performance. However, it may impose higher computational costs during inference. The authors could provide a more detailed analysis of the trade-offs between performance and inference efficiency, especially when using MVG with heavy medical files, like the high resolution MRI.

The ablation study lacks detailed insights into the contribution of individual components. For example, how do masked image modeling and autoregressive training individually affect the performance? For example, as the authors mentioned, usual augmentation techniques for SSL, like image cropping, can not be applied to medical imaging. The cropping procedure might mislead the model training and plumet the performance. I wonder if the authors addressed that and conducted some experiments with (specific for medical domain) data augmentation?

The paper compares the authors model with generalist models, however the comparative analysis with specialist models such as U-Net and nnUNet could be more detailed. A deeper examination of the performance trade-offs between generalist and specialist models would help to position MVG’s contribution more clearly and to understand the limits of its applicability. Also, I suggest addressing not only the segmentation task in a great detail; additional qualitative examples for inpainting and denoising would improve the clarity and show how MVG compares visually to specialists.

**Questions:**

Line 146 and subsection 4.2 :  “13 tasks…”.
I suppose it is a miswriting, since there are 13 datasets, but only 4 tasks.

Line 246 and Figure 3:
The sub-numeration of blocks in the figure would make the reference to its blocks and understanding easier. Instead of referring to blocks as "upper right" or "lower left".

Line 248: There is a reference to “Fig 3(a)”, but, again, no numeration in the image.

Table 1. I think it is important to note in the beginning of the article, that in the benchmark the proportion of datasets for 4 task in non-equal, with Segmentation task comprising the majority.

Line 255: “We hypothesize that this may be attributed to the masking strategy..." - Indeed, medical images data have fundamental and well known differences from the natural domain data; here, you could cite some papers that discuss this topic in more detail. For example (Huang et al., Self-supervised learning for medical image classification: a systematic review and implementation guidelines)

Line 465 and Table 4. The difference between Liver and Spleen datasets and Lung dataset is almost two times. Can you elaborate on the causes of such a huge difference?

Line 470: Data scalability section - there is no reference to Fig. 6.

The reliance on prompts to condition predictions introduces variability in outputs and performance, because different prompts may yield different results. Did authors address it somehow in their experiments? At the very least, how is the performance affected by the different prompts?

---

> ### Comment · Reviewer_Q4Bs · 2024-11-26
>
> There appears to be no rebuttal?

---

> ### Author Response · Authors · 2024-11-27
> **Rebuttal by authors -- Part I**
>
> Sorry for posting our rebuttal late. We address your concerns below:
>
> Q1: As shown by the authors, the hybrid use of autoregressive training boosts performance. However, it may impose higher computational costs during inference. The authors could provide a more detailed analysis of the trade-offs between performance and inference efficiency, especially when using MVG with heavy medical files, like high-resolution MRI.
>
>
>
>
> | Method            | GPU hours (RTX A5000) | mIoU |
> |--------------------|-----------------------|------|
> | Mask Image Modeling| 972h                 | 0.53 |
> | Autoregressive     | 1062h                | 0.79 |
>
>
>
> A1: We appreciate the question and apologize for any confusion caused. It is important to clarify that autoregressive training impacts only the training budget and does not introduce any additional computational costs during the inference stage. The observed higher computational costs arise primarily from two factors: the significantly larger input dimensions (our framework uses inputs that are four times larger, including the prompt image/label and task image) during training and the use of a larger network (ViT-Large) compared to standard segmentation methods, such as UNet.
> To provide further insight into how autoregressive training and mask image modeling affect the training budget, we conducted an ablation study (refer to the table above). This demonstrates that mask image modeling contributes only a marginal reduction in computational cost, as masking is applied solely to the label images. Autoregressive training, in contrast, slightly increases the training cost while significantly boosting performance (mIoU). It is worth emphasizing that neither method increases the computational burden during inference.
>
> Q2(1):The ablation study lacks detailed insights into the contribution of individual components. For example, how do masked image modeling and autoregressive training individually affect the performance?
>
> A2(1):
> Thank you for raising this point. We conducted an ablation study to evaluate the individual contribution of masked image modeling (MIM) and autoregressive (AR) training. As detailed in Table 7 of the original manuscript, AR training significantly outperforms MIM (using the optimal mask ratio of 75%) for segmentation tasks across all nine datasets, achieving an average improvement of 0.27 IoU across various organs and targets.
> As shown in line in Lines 254-263 in the original manuscript, we hypothesize that MIM’s suboptimal performance in segmentation tasks stems from its masking strategy, which may compromise the preservation of global contextual information critical for delineating anatomical structures, such as the spatial relationships among abdominal organs. This observation aligns with findings from [1], which indicate MIM excels at capturing local details but struggles with global context preservation. We also visualized attention maps and calculated attention distances. Our results, to be included in the revised manuscript, show that MIM-trained models exhibit smaller attention distances, indicating a focus on local regions, whereas autoregressive training captures more global context by producing larger attention distances in key attention heads.
> Conversely, MIM is beneficial for tasks like inpainting and denoising, where refining local details may take precedence over maintaining global context. To further validate this, we compared MIM and AR training for cross-modal synthesis, inpainting, and denoising tasks using mean absolute distance as the evaluation metric, as shown in the Table below. MIM consistently outperformed AR training in these tasks while also requiring less computational cost. Based on these insights and the results in Table 7, we adopted a hybrid training strategy:
> For segmentation tasks, we use AR training exclusively.
> For all other tasks, we allocate 90% of training iterations to MIM and 10% to AR training.
> | Method          | GPU hours (RTX A5000) | Cross-modal synthesis | Inpainting | Denoise |
> |------------------|-----------------------|------------------------|------------|---------|
> | MIM|972h|0.019|0.006|0.018|
> |             Autoregressive   | 1062h                | 0.020                  | 0.006      | 0.019   |

---

> > ### Author Response · Authors · 2024-11-27
> > **Rebuttal by authors -- part Ⅱ**
> >
> > Q2(2): For example, as the authors mentioned, usual augmentation techniques for SSL, like image cropping, can not be applied to medical imaging. The cropping procedure might mislead the model training and plummet the performance. I wonder if the authors addressed that and conducted some experiments with (specific for medical domain) data augmentation?
> >
> > A2(2): Thank you for your insightful comment. Our framework is designed to handle various medical imaging tasks—such as cross-modal synthesis, image segmentation, denoising, and inpainting—within a unified image-to-image generation approach. While task-specific data augmentation techniques can vary significantly across different medical datasets, we aimed to prioritize simplicity and general applicability. Following Painter, we adopted random cropping as our data augmentation strategy and found it works effectively across different datasets.
> > However, we acknowledge that random cropping may not always be ideal for certain medical imaging scenarios. Addressing this limitation is beyond the scope of our current work, as data augmentation is not the primary focus of this paper. In future research, we intend to conduct a systematic analysis of augmentation techniques tailored to the medical domain. This will enable us to identify and formalize a set of universal data augmentation strategies that can further enhance the generalizability and performance of our framework across diverse medical imaging tasks.
> >
> > Q3 (1): The paper compares the authors model with generalist models, however the comparative analysis with specialist models such as U-Net and nnUNet could be more detailed. A deeper examination of the performance trade-offs between generalist and specialist models would help to position MVG’s contribution more clearly and to understand the limits of its applicability.
> >
> > A3(1): Our pipeline unlocks the possibility of integrating multiple medical vision tasks into a unified image-to-image generation framework. While generalist models may not yet outperform domain-specific models like U-Net or nnUNet in segmentation, they offer greater flexibility. MVG, unlike specialist models, can be directly trained across diverse tasks and adapted to new datasets with minimal task-specific samples, whereas models like U-Net and nnUNet are limited to segmentation and cannot handle other tasks.
> > To position MVG’s contributions more clearly, we emphasize the following key points:
> > Vision-Centric Generalist Model:
> > MVG represents the first vision-centric medical generalist model capable of unifying diverse tasks across multiple modalities. While existing medical generalist models, such as MedSAM and UniverSeg, focus exclusively on segmentation, MVG extends this paradigm by incorporating different types of tasks like synthesis and denoising. This study provides the first proof-of-concept for the feasibility of a generalist medical AI model capable of performing a range of medical imaging tasks within a unified framework.
> > Hybrid Training Strategy:
> > MVG employs a hybrid training strategy that combines masked image modeling with autoregressive training, enhancing its conditional image generation capabilities and demonstrating strong scalability across both data and task levels.
> > Comprehensive Benchmark and Open-Source Contribution:
> > We introduce the first comprehensive medical imaging benchmark encompassing a  range of modalities, datasets, and anatomical regions. To facilitate further research and development, we will make all code and benchmarks publicly available to encourage future research on this new direction.
> >
> > Q3(2)Also, I suggest addressing the segmentation task in great detail; additional qualitative examples for inpainting and denoising would improve the clarity and show how MVG compares visually to specialists.
> >
> > A3(2): Thanks for the suggestion. We have provided qualitative examples of inpainting and denoising denoise (LowDose), inpainting (BrainLocal), and cross-modal synthesis (BrainGLI)  in Figure 5 of our original manuscript. We will provide more qualitative examples in the next version.
> >
> > Q4: Line 146 and subsection 4.2 : “13 tasks…”. I suppose it is a miswriting, since there are 13 datasets, but only 4 tasks.
> >
> > A4: Thank you for bringing this to our attention! Yes, this was a miswriting. There are 13 datasets and 4 tasks, not 13 tasks. We will revise the manuscript in the next version.
> >
> > Q5: Line 246 and Figure 3: The sub-numeration of blocks in the figure would make the reference to its blocks and understanding easier. Instead of referring to blocks as "upper right" or "lower left".
> > Line 248: There is a reference to “Fig 3(a)”, but, again, no numeration in the image
> >
> > A5: We will update the references from "(a)/(b)" to “left/right”

---

> ### Author Response · Authors · 2024-11-27
> **Rebuttal by authors -- Part Ⅲ**
>
> Q6: Table 1. It is important to note at the beginning of the article that in the benchmark the proportion of datasets for 4 tasks in non-equal, with the Segmentation task comprising the majority.
>
> A6: We acknowledge that the number of segmentation datasets in the benchmark is higher compared to other tasks. However, the total size of the data used for low-level tasks (e.g., denoising and inpainting) is substantially larger. Specifically, the benchmark consists of 2,475,636 training image/label pairs, distributed as follows:
> Segmentation: 212,811 pairs
> Synthesis: 1,987,111 pairs
> Inpainting: 139,311 pairs
> Detection: 66,532 pairs
> Denoising: 69,871 pairs
> To ensure a balanced contribution from high-level and low-level tasks, we adjusted the sampling weights. The segmentation task is assigned a sampling weight of 0.5, while all low-level tasks collectively share the remaining 0.5, to help maintain a more equitable representation of the tasks during training.
>
> [1] Xie, Z., Geng, Z., Hu, J., Zhang, Z., Hu, H., & Cao, Y. (2023). Revealing the dark secrets of masked image modeling. In Proceedings of the IEEE/CVF conference on computer vision and pattern recognition (pp. 14475-14485).

---

> > ### Author Response · Authors · 2024-11-30
> >
> > Dear Reviewer Q4Bs,
> >
> > We sincerely appreciate your review. We have carefully considered each of your questions and provided detailed responses in the rebuttal. Please let us know if you have any further questions or concerns.
> >
> > Thanks!

---

> ### Author Response · Authors · 2024-12-01
>
> Dear Reviewer Q4Bs,
>
> We would like to kindly remind you that the discussion deadline is approaching. After this deadline, we may not have the opportunity to respond.
>
> We sincerely appreciate your time and thoughtful input and look forward to hearing your feedback.
>
> Sincerely,
>
> Authors of Submission 8515

---

### Note · Authors · 2024-12-03

I have read and agree with the venue's withdrawal policy on behalf of myself and my co-authors.